# The impact of official recommendations during the COVID-19 pandemic on the clinical activity and business turnover of manual therapists in Sweden–The CAMP cohort study

Emmanuel Aboagye[1,2]*, Eva Skillgate[2,3,4], Nathan Weiss[2,3], Iben Axén[2,4,5]

1 Department of Psychology, Norwegian University of Science and Technology, Trondheim, Norway, 2 Unit of Intervention and Implementation Research for Worker Health, Institute of Environmental Medicine, Karolinska Institutet, Stockholm, Sweden, 3 Department of Health Promotion Science, Musculoskeletal and Sports Injury Epidemiology Center, Sophiahemmet University, Stockholm, Sweden, 4 Naprapathögskolan-Scandinavian College of Naprapathic Manual Medicine, Stockholm, Sweden, 5 The Norwegian Chiropractic Research Foundation, ELIB, Oslo, Norway

* emmanuel.aboagye@ntnu.no

## Abstract

### Background

This study examined manual therapy business owners' perception of official recommendations during the COVID-19 pandemic and the impact on their clinics' economic performance, including clinic activity hours and business turnover.

### Materials and methods

In a longitudinal study design, data were collected in November 2021 (baseline), and after three months, six months, and 12 months. Participants were manual therapists who were business owners. A growth curve model was used to analyze differences in clinical activity trajectories. Multinomial logistic regression analysis was used to assess the relationship between perceived disruptions in business and turnover. Qualitative text analysis was used to examine participants' responses to open-ended questions concerning economic measures taken to sustain their business during the pandemic.

### Results

This study of 443 manual therapy business owners found that clinics were initially active with minimal variation, but activity changed following COVID-19 recommendations. Business owners perceived that the disruptions had no significant impact on turnover during the initial stages of the official recommendations. Economic support and the previous decrease in turnover increased the likelihood of experiencing a decreased turnover at 12 months. Business owners implemented cost-cutting measures and diversified income sources to navigate COVID-19 challenges and sustain their businesses.

**Data Availability Statement:** Due to ethical restrictions of disclosing personal data, authors have to seek permission to allow us to make the

data used in this study available. Data will be available upon request after permission is granted from the Karolinska Institutet's Ethics Review Board in Stockholm whose contact is kansli@stockholm.epn.se. Inquiries for data access should first be sent to iben.axen@ki.se, who will then contact the ethics board for permission to openly share the data.

**Funding:** The research was funded by grant number 200140 from AFA Insurance and The Swedish Naprapathic Association. The funders had no role in study design, data collection and analysis, decision to publish, or preparation of the manuscript.

**Competing interests:** The authors have declared that no competing interests exist.

## Conclusion

The official recommendations in Sweden had an impact on manual therapists' businesses as the COVID-19 pandemic lingered. Some business owners were concerned at the early stages about lower turnover but showed financial resilience by cutting costs and finding new revenue sources to overcome COVID-19 challenges.

## Introduction

The Coronavirus disease-2019 (COVID-19) was discovered in late 2019. By early March 2020, the disease spread rapidly, and the World Health Organization (WHO) declared a pandemic [1]. From there on confirmed cases and deaths due to COVID-19 globally were cumulatively tracked [2]. With an estimated global average fatality rate of 3%, COVID-19 had a high observed case-fatality ratio in some countries such as UK and USA. The daily confirmed COVID-19 deaths per million people had reached as high as 13% by January 2021 for all countries affected [2]. In Sweden, by 1 September 2020, 0.8% of Swedish residents had tested positive for the virus and 0.06% of the population had died, which was higher than the neighboring Nordic countries [3]. During this period, Sweden experienced a COVID-19 infection rate approximately three times higher and a mortality rate nearly nine times higher compared to the average in the neighboring Nordic countries, with the exception of Finland, which had a significantly higher infection rate [2].

The COVID-19 pandemic caused unprecedented health and economic challenges. Regulations for physical distancing and other measures to control the spread of the virus were enforced in many countries, disrupting work and social life. These changes, while considered necessary responses to the pandemic in almost all countries worldwide, threatened even the operations of public and private healthcare, especially patient-facing types of health care. Small businesses encountered significant challenges because of official recommendations aimed at preventing the spread of COVID-19 [4].

Sweden chose a different disease prevention and control path during the pandemic than many other European countries [5]. Its approach to COVID-19 differed from many other countries, as it did not enforce nationwide lockdowns and imposed fewer and less stringent restrictions. The response involved implementing various measures throughout the pandemic to control the virus's spread and lessen its impact on healthcare, individuals, and businesses. Sweden's strategy focused on non-binding restrictions, emphasizing personal responsibility through practices like social distancing, hand hygiene, and self-isolation when experiencing mild symptoms [6]. Additional guidelines included remote work, virtual schooling, essential-travel-only policies, and limitations on public gatherings. The main authority overseeing Sweden's COVID-19 protocols was The Public Health Agency of Sweden, with regional directives also in place to manage local outbreaks, if needed [5]. Sweden's overall excess mortality from 2020 to 2021 was 0.79%, lower than in many other European countries [7]. This outcome was attributed to Sweden's strategy and control policy. In addition to voluntary recommendations, Sweden implemented a wide range of temporary changes in its social protection and inclusion schemes, primarily for small businesses to ensure service continuity, and measures to address deficiencies in long-term care and healthcare [8].

Manual therapists, as licensed chiropractors and naprapaths, treat musculoskeletal pain and disability using manual therapy The traditional practice is that these therapists work in close physical proximity to the patient since they use their hands as a part of their assessment and

treatment for managing patients' pain and disability [9]. An increasing number of people seek care from these therapists for common musculoskeletal conditions, such as back and neck pain [10]. However, there were challenging circumstances in the context of unanticipated governmental recommendations on social distancing and public health responses.

The rapid evolution of the COVID-19 pandemic led to sharp reductions in elective visits by patients due to the recommendations (Moore 2021). It was anticipated after the few months into the official recommendation were in place that nearly all operational small clinics would encounter financial challenges [4]. Many of these small clinics were expected to handle significant financial shocks and lack the resources needed to adequately respond to the pandemic [11]. For example, financially vulnerable businesses may have limited ability to invest in COVID-19 response measures to remain operational. These threats to their financial stability would indicate potential negative effects on the business economy.

Also, through the social protection and inclusion schemes, some small businesses were promised lump-sum governmental money as a buffer to mitigate the decreased demand and shortfalls in finances during the pandemic restrictions [8]. However, there is limited evidence regarding the effects of COVID-19-related official recommendations and economic measures implemented to ensure the survival of businesses [12]. This information is relevant for research, policy, and practice as future business behaviors are likely to change with new information relating to modified government and local restrictions.

Hence, it is crucial to investigate how owners of chiropractic and naprapathy practices perceived the impact of official recommendations and revenue sources on their businesses, as these factors could also influence the survival of small but thriving clinics. From this viewpoint, the purpose of this study was to examine the perception of business owners towards official recommendations during the COVID-19 pandemic and their impact on the economic performance of licensed manual therapy clinics, specifically in terms of clinic activity hours and business turnover.

## Materials and methods

The study presents findings from a nationwide prospective cohort study called 'Corona and Manual Professions' (CAMP) research project (Clinical Trials register identifier: NCT04834583). Its primary aim was to examine the impact of the COVID-19 pandemic on the health, work environment, and economic aspects of licensed Swedish manual therapists (i.e., chiropractors and naprapaths) over a 12-month period, starting at the second wave of the pandemic in Sweden, in November 2020. The Swedish Ethical Review Authority approved the research with Dnr: 2020–03836. Refer to the S1 Checklist.

### Study design

A longitudinal cohort study design was used to examine the perceived business disruption resulting from official recommendations. In this approach, repeated data collection was conducted four time points over a 12-month period, allowing for the analysis of changes in perceived business disruption and its impact on the clinical activity hours and business turnover.

### Study sample

Participants in the study were recruited from a national register of licensed manual therapists in Sweden, as described previously [13]. The recruitment process involved providing a generic link to access information about the study and the web-based questionnaire. The following criteria were applied to identify eligible participants for inclusion in the analysis. First, participants had to be clinically active and practicing in Sweden as the study examines the effects of

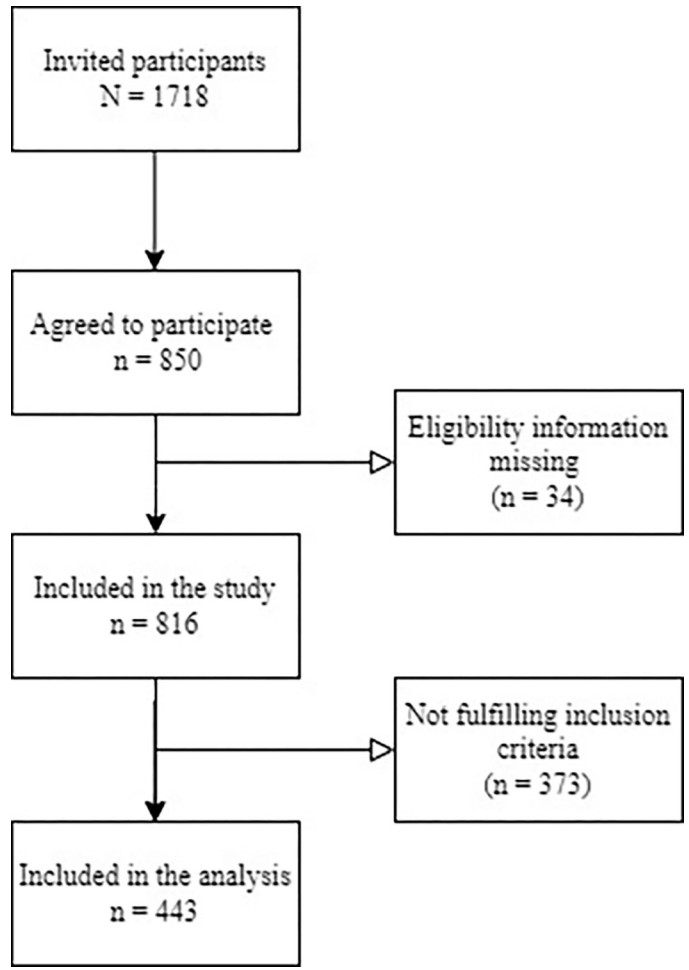

**Fig 1. Study participant flow chart.** The flow chart showing the number of invited participants, excluded participants, and included participants in the study and analyses.

COVID-19 official recommendations in this country. Second, eligible participants were required to hold a professional certification in either chiropractic or naprapathy. Lastly, participants were expected to be an owner (solo or joint) of their clinic. In the CAMP cohort, 1718 participants were invited, and 850 agreed to participate. Thirty-four participants were excluded for not providing eligibility information, resulting in a study sample of 816, with a response rate of 47%. After applying the inclusion criteria described above, 443 were included in the final sample for this study (Fig 1).

## Data collection

The CAMP study collected data through four web surveys over a year. Written informed consent was obtained from the participants of the study. Data were collected at baseline in November of 2020, after three months, six months, and 12 months. REDCap, an electronic data tool available through the Karolinska Institutet, was used to manage data collection [14]. The survey included questions on the following domains: 1) Demographic factors; 2) Profession-related factors; 3) Work environment; 4) Health-related factors; 5) Lifestyle factors; 6) Psychological factors; 7) Business economy. This study examines questions in the domain of business economy.

### Perceived business disruptions due to official recommendations

Participants were asked to comment on the following statement: "Official recommendations have interfered with clinical practice", with the answer alternatives: "Not at all", "Yes, to some degree", "Yes, to a moderate degree", "Yes, to a large degree", and "I have not been clinically active during the COVID-19 pandemic". The variable was later dichotomized by categorizing all the "yes-responses" into 1 = "yes" and "Not at all" to 0 = "no".

### Clinical activity

We evaluated the clinical activity by analyzing their hours of operation in the past three months. The surveys requested the practitioners to report the average number of hours they were clinically active per week during this period.

### Economic consequences due to the pandemic

Business owners were asked about the actual revenue for 2020 compared to 2019 at 12 months. The economic performance of clinics was categorized as "Decreased", "Same as 2019", and "Increased" turnover in 2020 at 12 months.

### Economic measures and sources of revenue

To describe what measures had promoted a sustainable business economy, the questionnaire also asked what measures and sources of revenue may have influenced the survival of the clinics in connection with COVID-19. An open response question asked "what measures have you taken in connection with COVID-19 to promote a sustainable economy in your business?" Participants provided free text responses to the question on the nature of the measures taken and other sources of income accessed to keep the business during the COVID-19 when business had slowed down.

### Analyses

Estimates were computed using descriptive statistics, which provide valuable information about the characteristics and experiences of chiropractic and naprapathy practices. The estimates also provide information on the prevalence of perceived business disruption caused by official recommendations at various time points, including baseline, three months, six months, and 12 months. For continuous variables such as age and clinical activity in hours, the mean and standard deviation statistic was reported indicating the central tendency and dispersion of attributes. Gender, clinic focus, business turnover, and other categorical attributes were described using frequencies and proportions. The statistical analysis of the collected data was conducted using the SPSS and R package Lavaan [15, 16]. To assess the significance of the findings, the conventional criterion of statistical significance at the 0.05 level was applied to all conducted tests.

The analysis began by creating line graphs and randomly selecting approximately 10% n = 44 business owners to examine the operating hours over time and assess the variability within the chosen sample. A growth curve model with a maximum likelihood (ML) estimator was used to explore the potential differences in slopes of the dependent variable, clinical activity [17]. The ML estimator, which considers potential non-normal distribution and heteroscedasticity (error variance) in the data, is particularly well suited for analyzing longitudinal data with repeated measures because it provides robust estimates of the model parameters [18]. Intercept-only model and a linear growth model were used to examine potential significant variations in the rate of change of clinical activity over time within these businesses. A multi-group growth model with group categories "yes" or "no" was used to compare the clinical

activity of business owners who felt their businesses were disrupted by COVID-19 with those who didn't feel disrupted, over a period of 12 months. This helped us understand if and how their clinical activities differed based on their perceptions of disruption.

To examine the association between perceived business disruptions due to official recommendations during the pandemic and the economic performance of businesses owned by manual therapists, a multinomial logistic regression was used [19]. The goal of using this method was to estimate an occurrence probability model of an event based on the maximum likelihood method. The method accounts for the categorical nature of the dependent variable since the economic performance of clinics were categorized into three nominal outcome categories: "Decreased", "Same as 2019", and "Increased" turnover in 2020 at 12 months. Multinomial regression allows for the assessment of the probability of being in each outcome category relative to a reference category while controlling for relevant influencing variables.

To provide context for closed-question survey responses, the free-text responses from manual therapists were analyzed regarding the steps they took to support a sustainable economy in their businesses during the COVID-19 pandemic. The free-text responses included individual words or phrases which were gathered and compiled into a structured file to enable summative content analysis [20]. Most responses consisted of keywords and did not require any interpretation. The content of the free text replies was analyzed by categorizing closely related replies under key terms. The analysis focused on identifying and classifying the cost-cutting or optimization strategies and revenue sources mentioned in the responses to assess how the various actions may have contributed to the business survival during the pandemic.

## Results

### Characteristics of participants

Based on a sample size of 443, Table 1 describes the characteristics of manual therapists as business owners. The average age was 47 years. The gender distribution was nearly equal, with 53% males and 47% females. The clinical activity took up 28 hours per week on average over the last three months. In terms of clinic focus, 52% owned clinics with exclusively chiropractic or naprapathy, while 48% owned clinics with a mix of professions represented. According to the ownership relationships, 53% owned their clinics, 29% were co-owners, 15% were self-employed in someone else's clinic. In terms of turnover expectation at baseline in November 2020, 57% reported an expected decrease, 16% expected no change, and 21% reported an expected increase from 2019, the year before the COVID-19 pandemic. The table also shows that business owners perceived some level of business disruption due to official recommendations at various time points. At baseline, 84% reported disruptions, which decreased to 68% at three months, 56% at six months, and 49% at 12 months. Thus, the majority experienced business disruptions because of official recommendations across all time points.

### Changes in clinical activity

The clinical activities of the included manual therapist business owners were measured in hours of operation over four time points, as shown in Figs 2 and 3. The line graphs for the first selected sample of business owners (n = 44, 10% and n = 111, 25% of the total sample) in the study showed relatively flat lines, indicating little to no variability in clinical activity. Figs 2 and 3 indicates that the clinical activity of most of the business owners surveyed did not change due to the official recommendations regarding the COVID-19 pandemic.

Using a growth curve model, we further examined whether the slopes of the dependent variable clinical activity (hours of operation per week in the past 3 months) were significantly different from zero at baseline measurement for chiropractic and naprapathy practices in a no

**Table 1. Characteristics of manual therapists owning chiropractic and naprapathy practices (n = 443).**

| Attribute | Definition or categories | Mean (SD) / numbers (%) |
|---|---|---|
| Age | Age | 47 (10) |
| Gender | Males | 211 (53%) |
|  | Females | 185 (47%) |
| Average clinical activity [a] | Hours active per week (past 3 months) |  |
| At baseline | Clinical activity at baseline | 28 (11) |
| At 3months | Clinical activity at 3 months | 29 (16) |
| At 6 months | Clinical activity at 6 months | 29 (11) |
| At 12 months | Clinical activity at 12 months | 30 (11) |
| Clinic composition [b] | Only chiropractic/naprapathy | 222 (50%) |
|  | Several different professions | 209 (47%) |
|  | Primary care rehab | 7 (2%) |
| Ownership relations [c] | Own my clinic | 232 (52%) |
|  | Co-owner of clinic | 127 (29%) |
|  | Self-employed in clinic | 68 (15%) |
| Turnover year 2020 [d] | Expected decreased turnover | 254 (57%) |
|  | Same as 2019 | 72 (16%) |
|  | Expected increased turnover | 92 (21%) |
| Business disruptions [e] |  |  |
| at baseline [Yes] | Business disruption at baseline | 374 (84%) |
| at 3 months [Yes] | Business disruption at 3 months | 301 (68%) |
| at 6 months [Yes] | Business disruption at 6 months | 246 (56%) |
| at 12 months [Yes] | Business disruption at 12 months | 219 (49%) |

[a]Clinical activity described in terms of hours per week on average in the past three months.

[b]Clinic composition was classified as chiropractic or naprapathy, or a combination of other professions.

[c]Ownership relationships are classified as owning their clinics, co-owned their clinics, self-employed in someone else's clinic.

[d]Business turnover in 2020 is divided into three categories: decrease, no change, and increase from 2019, the year before the COVID-19 pandemic.

[e]The perceived level of business disruption caused by official distancing recommendations at various time points is defined as disruption at baseline, 3m, 6m, and 12m.

growth model (i.e., no predictors). The results show that the mean estimate of clinical activity (mean = 29 hours, p < .001) was significantly different from zero at baseline. The model chi-square test was significant, suggesting that the model did not fit the data adequately. The data is characterized by a linear growth model. Results from the linear model indicate a good fit to the data, with CFI/TLI = 0.99; $\chi^2$ = 6.48, df = 5, p-value = 0.263; SRMR = 0.03. The intercept and slope factors have mean values of 28 hours and 0.89, respectively, with p < .001. The results suggest a significant change in clinical activity over time, with an initial activity of 28 hours and an average change of 0.89 per unit of time (i.e., in the past three months).

The invariant test in the multigroup growth model with growth factor means freely estimated, variances and covariances held equal was used to analyze differences in business owners' perceptions of disruptions at 12 months and thus affecting clinical activity. Business owners perceiving disruption reported lower baseline clinical activity (mean = 26 hours, p < .001) with an average change of 0.89 per unit of time over the past three months. Conversely, business owners perceiving no disruption reported higher baseline clinical activity (mean = 30 hours, p < .001) with an average change of 0.78 per unit of time over the same period. This

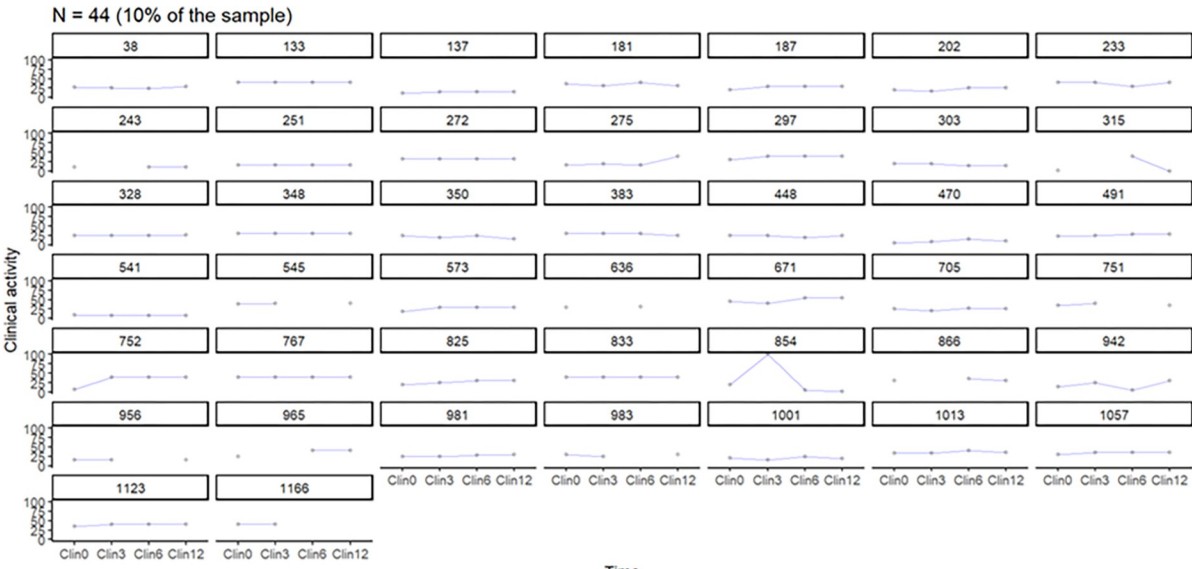

**Fig 2. The clinical activities of chiropractic and naprapathy practices, n = 44.** The y-axis represents clinical activity measures in hours (per 3 months), while the x-axis represents the four time points (referred to as time).

indicates that the rate of change for those perceiving disruption was more pronounced than for those without disruption. The Chi-Squared Difference Test indicated that the model with freely estimated means was a good fit to the data, suggesting a difference in clinical activity over time between disrupted and non-disrupted businesses as per official recommendations.

## Business turnover

A multinomial regression was used to analyze predictors for an ordinal group classification, i.e., the actual change in business turnover which decreased compared to the year prior the pandemic i.e., 2019, business turnover which stayed the same in this period, and business turnover which increased in this period. The reference category for the outcome variable was 'business turnover which increased'. Each of the other two categories was compared to this reference group.

Table 2 shows parameter estimates (model coefficients) investigating the association between perceived COVID-19-related business disruptions due to the official recommendations and actual business turnover (at 12-month follow-up). This analysis considers the previous three to six-months turnover and government economic support. The inclusion of these predictors enhanced the fit between the model and the data, as indicated by the $\chi^2$ test (df = 18, n = 313, $\chi^2$ = 91.5,

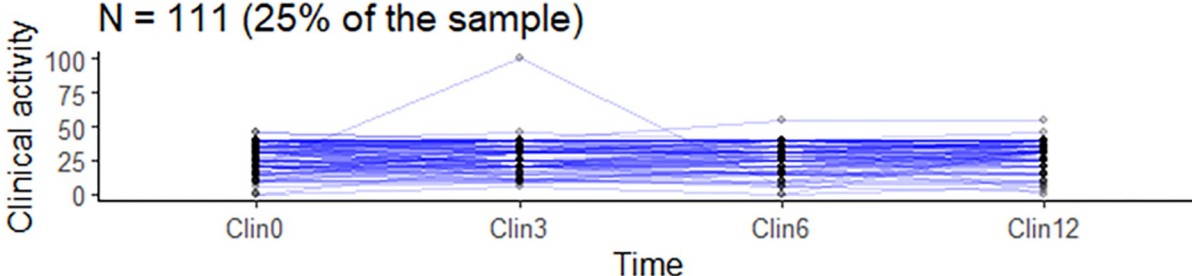

**Fig 3. The clinical activities of chiropractic and naprapathy practices, n = 111.** The y-axis represents clinical activity measures in hours (per 3 months), while the x-axis represents the four time points (referred to as time).

**Table 2. The odds ratio (OR) of predictors related to the turnover of business owners operating chiropractic and naprapathy practices.** (n = 313).

| Predictor | Decreased turnover | | | Same turnover as 2019 | | |
|---|---|---|---|---|---|---|
| | OR | SE | 95% CI | OR | SE | 95% CI |
| Clinical activity at baseline | .95* | .02 | .93 - .99 | .99 | .02 | .96–1.03 |
| Disruptions at 3m = [No] | 1.28 | .39 | .59–2.76 | 1.87 | .42 | .83–4.23 |
| Disruptions at 3m = [Yes] | . | . | | . | . | . |
| Disruptions at 6m = [No] | .55 | .36 | .27–1.12 | .91 | .42 | .40–2.05 |
| Disruptions at 6m = [Yes] | . | . | | . | . | . |
| Disruptions at 12m = [No] | .45* | .34 | .23 - .87 | 1.09 | .42 | .48–2.46 |
| Disruptions at 12m = [Yes] | . | . | | . | . | . |
| Economic support = [Yes] | 2.09* | .32 | 1.12–3.89 | 1.92 | .37 | .93–3.94 |
| Economic support = [No] | . | . | | . | . | . |
| Turnover at 3m = [Decreased] | 3.39** | .36 | 1.67–6.85 | 1.39 | .44 | .59–3.26 |
| Turnover at 3m = [Same] | 3.44* | .53 | 1.23–9.63 | 1.30 | .57 | .42–4.01 |
| Turnover at 3m = [Increased] | . | . | | . | . | . |
| Turnover at 6m = [Decreased] | 1.51 | .36 | .74–3.06 | 1.02 | .48 | .39–2.62 |
| Turnover at 6m = [Same] | .95 | .51 | .348–2.57 | 3.29* | .50 | 1.23–8.79 |
| Turnover at 6m = [Increased] | . | . | | . | . | . |

Disruptions = perceived disruptions in business due to COVID-19 official recommendations, controlling for clinical at baseline, economic support from the government and turnover from the previous months.

OR = odds ratio

SE = standard error

*p < .05

**p < .01

Nagelkerke $R^2$ = 0.31, p < .001). Engaging in clinical activity in the past three months, perceived disruptions in business due to official recommendations at month 12, government economic support, and the level of business turnover in the past three and six months compared to the 2019 annual turnover all made significant unique contributions. The goodness-of-fit test did not yield a significant result indicating that the model fits the data well.

The results in Table 2 suggest that clinical activity at baseline is associated with lower odds of decreased turnover (OR = 0.95, 95% CI = 0.93–0.99). There is no significant association between clinical activity at baseline and turnover remaining the same as in 2019. Perceived business disruptions at 3m and 6m are not individually associated with turnover changes.

However, participants responding no perceived disruptions at 12m was associated with a lower odd of decreased turnover (OR = 0.45, 95% CI = 0.23–0.87). In relation to the economic situation of the business, practices receiving government economic support had higher odds of decreased turnover (OR = 2.09, 95% CI = 1.12–3.89). There was no significant association between the absence of economic support and turnover changes. Further, the results show that practices experiencing decreased turnover at 3m had significantly higher odds of decreased turnover (OR = 3.39, 95% CI = 1.67–6.85). Practices with increased turnover at 6m had higher odds of increased turnover (OR = 3.29, 95% CI = 1.23–8.79).

## Economic measures and sources of revenue to sustain business (responses to open-ended questions)

The free-text responses provided by business owners were analyzed to understand the range of strategies adopted aimed at maintaining financial stability in the face of COVID-19 economic challenges. These answers were examined, and words and phrases were extracted. The

economic measures taken included various actions such as cost reduction efforts, measures related to rent and facilities, salary and work hours management, and employment-related actions. Participants chose words and phrases such as minimizing expenses, cutting down on unnecessary costs, scaling back on investments, negotiating with landlords for reduced rent or obtaining rent concessions, taking lower pay, preferring to work more, postponing planned renovations, postponing new hires, and reducing personnel costs through layoffs.

There were also instances of businesses demonstrating a proactive approach to adapt and diversify income sources. The categories that could be classified as revenue sources included: adaptations in service offerings, applying for various forms of financial support (including government grants and wage subsidy programs), changes related to financial planning, savings, and loans, communication, and transparency with patients regarding adherence to official guidelines and safety protocols, and commitment to adapting marketing tactics and business strategies to drive revenue. Participants chose words and phrases such as taking on new roles in different jobs, modifying their clinics' services, taking loans or obtaining loan repayment deferments or adjustments, offering remote support to clients through videos or online sessions, implementing hygiene-related measures in the work environment, participating in online networking events, and creating various online marketing campaigns.

## Discussion

The COVID-19 pandemic brought about social-distancing guidelines and economic changes that raised concern about the survival of small businesses. This study sought to shed light on the impact of the Swedish voluntary recommendations by exploring how they affected the operations and financial outcomes of manual therapist practices. This is to inform policy designed to support active business owners and to better know where extra resources should be targeted to those affected by a pandemic.

When examining the trajectories in clinical activity over time among businesses, the findings show that the mean estimate of clinical activity at baseline significantly differed from zero. This indicates that clinics were actively providing services at baseline, with minimal variability in clinical activity due to the official recommendations in place during the COVID-19 pandemic. The linear model used showed variation in clinical activity over time. This finding highlights the dynamic nature of clinical activity of businesses and the need to consider temporal trends when evaluating clinic performance and allocating resources. Interestingly, business owners perceiving disruptions reported lower baseline clinical activity compared to those perceiving no disruption, indicating an initial disparity in activity levels between the two groups. This suggests that the presence of disruptions may have influenced the operational capacity of clinics even before the 12-month, possibly due to the challenges posed by adhering to official recommendations amidst the pandemic.

The turnover of business owners operating chiropractic and naprapathy practices during the COVID-19 pandemic was investigated. Firstly, clinical activity at baseline showed a significant association with turnover. This suggests that businesses with higher clinical activity at baseline is associated with slightly lower odds of experiencing decreased turnover. Secondly, perceived disruptions in business due to COVID-19 recommendations at 12 months also emerged as significant predictors of turnover outcomes. This finding is consistent with previous studies showing that some businesses, particularly those experiencing long-lasting disruptions and early-stage losses in turnover, were hit especially hard [4, 11]. Further, economic support from the government was found to be positively associated with decreased turnover, suggesting that clinics receiving economic assistance were more likely to maintain or improve their turnover. Turnover in the preceding months was strongly associated with future turnover

outcomes. Clinics that experienced decreased turnover at 3 months were significantly more likely to continue experiencing decreased turnover at subsequent time points, emphasizing the persistence of financial challenges faced by these businesses. These predictors can help inform targeted initiatives and support measures to strengthen the resilience and sustainability of small healthcare businesses in future crises.

The variations in the clinical operations of these manual therapy practices during the COVID-19 pandemic were anticipated to be closely linked to their financial performance. For instance, businesses whose turnover remained the same at 6 months were less likely to see decreased turnover later or more likely to maintain 2019 turnover. This suggests that for some businesses stabilizing their revenue was crucial for bouncing back from initial challenges, which is why they may have required strategic adjustments. This finding is further supported by the findings of a previous study, indicating that small businesses consistently evaluated and adapted their strategies to with evolving circumstances during the pandemic to stay afloat [12]. This stability may reflect the adaptability and resilience of these business owners in adjusting their operations to adhere to official recommendations while still providing essential services to their patients. This emphasizes the findings in previous studies that show business owners recognized and abided by changing governmental regulations but maintained continuity in patient care even in the face of significant external disruptions [11, 12].

The study also investigated the economic measures business owners took that contributed to sustaining their businesses during the pandemic. The identified categorizations, such as cost reduction efforts, rent and facilities management, salary adjustments, and employment-related actions, provide a comprehensive framework for small businesses to adapt and respond effectively to the financial constraints imposed by a pandemic. This implies that the businesses managed to implement effective strategies to adapt and mitigate the potential effects of prolonged disruptions. Overall, these diverse actions demonstrate the resilience and adaptability of the business owners in navigating the challenges posed by the pandemic to ensure the continued survival of their clinics. The study findings provide a rich resource of practical strategies that seem to suggest that business owners used the perceived level of current business during difficult economic times to navigate the evolving landscape [21].

The business owners also showed proactivity in diversifying their income sources to mitigate the economic challenges posed by the pandemic. The highlighted categories related to revenue sources such as the adaptations in service offerings, and transparent communication with patients demonstrate innovative and proactive approaches to catering to evolving patient needs while expanding revenue streams. The commitment to adapting marketing tactics and business strategies aligns with the evolving landscape of patient preferences and helps drive revenue through innovative approaches such as online networking events and digital marketing campaigns. These results align with previous studies, such as Moore et al. [11], which indicated that practitioners have implemented significant COVID-19 infection control measures within their practice settings all aimed at safeguarding their businesses. Further, seeking financial support through government grants, wage subsidy programs, and taking loans or loan repayment adjustments allowed business owners to manage their cash flow effectively while maintaining operational continuity. This highlights the results of a prior study indicating that most professionals had to seek financial aid because of their income loss [11]. It also emphasizes the significance of utilizing accessible resources to maintain their businesses amidst these uncertain times.

The study is one of a few to investigate the economic impact of the COVID-19 policy restriction on businesses, and the strategies adopted by businesses and employers. Public health policies put in place during the COVID-19 pandemic have been assessed for their effectiveness in reducing transmission and minimizing health outcomes like death [22]. Limited

research discusses the association between COVID-19 policy restrictions and economic impact, though not thoroughly examined. A descriptive study across eleven countries showed how chiropractors used innovative strategies such as telehealth and outreach to communicate and care for patients [12]. A global survey found that most chiropractors (85%) followed regulatory advice by using telehealth, personal protective equipment, and other measures in their practice [11]. Economic support such as wage and business subsidies were also mentioned, which were accessible in certain countries like Canada and Sweden, to help businesses and employers cope with costs and job losses. The findings in the previous research are consistent with our findings in terms impact and strategies adopted by businesses and employers. This information could advise government bodies and professional associations that manual therapists are able to implement a multifaceted strategy to provide financial support initiatives during future pandemics. These efforts have the capability to empower business owners, strengthen their resilience, and foster growth in the face of ongoing challenges.

## Strengths and limitations

There are several strengths. It is a substantial study, although we cannot be certain that it is a representative sample. Another strength is the high follow-up rate (80% response rate in the longitudinal measurements) which perhaps minimizes the risk of selection bias. The high follow-up rate also provides confidence in the reliability of the data collected over time including the crucial time points investigated (period during the second, third and fourth wave of the COVID-19 pandemic). This allows for a more accurate understanding of any changes or trends observed in the study.

It is important to acknowledge that the surveys use validated questions from different tools, covering the same areas as the project's research questions. However, some questions were made specifically for the project to evaluate the financial conditions of manual therapy clinics or address aspects not covered by existing tools. These self-made questions have limitations and have not undergone rigorous validation like established tools, compromising their reliability and validity. Further, some original scales were dichotomized to simplify the analysis and interpretation but can be a source of classification challenge of the groups. However, the decision to dichotomize the variable is conceptually clear. For instance, all responses indicating any degree of interference with clinical practice ("Yes, to some degree", "Yes, to a moderate degree", "Yes, to a large degree") were grouped together as a "yes" response, indicating some level of disruption. The "Not at all" response was categorized as "no", indicating no disruption. All "yes" responses share the common characteristic of indicating disruption, which justifies treating them as a single group by focusing on the presence or absence of disruptions.

Researchers and policymakers should also be aware of the limitations associated with the study's temporal focus, scope, and geographic specificity when interpreting and applying the findings to different contexts. Firstly, the specific time points investigated over a one-year period restrict the generalizability of the findings, as the economic and healthcare landscape continued to evolve differently in different countries throughout the COVID-19 pandemic. Therefore, caution should be exercised when applying these results to current or future contexts. Secondly, the study investigated the perception of business disruptions due to official recommendations in relation to clinical activity and turnover. While this was the focus in the chosen context, it may not fully capture all relevant factors influencing business operations during the pandemic. Finally, the data for the study was gathered in the context of preventing COVID-19 infections in Sweden. As a result, the applicability of these findings to regions that used a different approach, policies, and faced different economic conditions should be approached with contextual variations in mind.

## Conclusion

The voluntary official recommendations in Sweden had an impact on the clinical activity of manual therapists with limited disruptions in business at the latter stages of the pandemic. Some businesses expressed concern about lower turnover but showed financial resilience by cutting costs and finding new revenue sources to overcome COVID-19 challenges.

## Supporting information

**S1 Checklist. TREND statement checklist.**
(PDF)

## Acknowledgments

The authors would like to express their gratitude to Sofie Jonsson for her project administration and assistance, as well as to the participating manual therapists for their valuable contribution.

## Author Contributions

**Conceptualization:** Eva Skillgate, Iben Axén.

**Data curation:** Emmanuel Aboagye, Eva Skillgate, Nathan Weiss, Iben Axén.

**Formal analysis:** Emmanuel Aboagye, Nathan Weiss, Iben Axén.

**Funding acquisition:** Eva Skillgate, Iben Axén.

**Investigation:** Eva Skillgate, Iben Axén.

**Methodology:** Emmanuel Aboagye, Eva Skillgate, Nathan Weiss, Iben Axén.

**Project administration:** Eva Skillgate, Iben Axén.

**Supervision:** Iben Axén.

**Writing – original draft:** Emmanuel Aboagye, Eva Skillgate, Nathan Weiss, Iben Axén.

**Writing – review & editing:** Emmanuel Aboagye, Eva Skillgate, Nathan Weiss, Iben Axén.

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
