## [Decision Letter · Decision Letter 0]

31 May 2024

PONE-D-24-08211The impact of official recommendations during the COVID-19 pandemic on the clinical activity and business turnover of manual therapists in Sweden – the CAMP cohort study.PLOS ONE

Dear Dr. Aboagye,

Thank you for submitting your manuscript to PLOS ONE. After careful consideration, we feel that it has merit but does not fully meet PLOS ONE’s publication criteria as it currently stands. Therefore, we invite you to submit a revised version of the manuscript that addresses the points raised during the review process.

We look forward to receiving your revised manuscript.

Kind regards,

Jenny Wilkinson, PhD

Academic Editor

PLOS ONE

Journal Requirements:

 [The research was funded by grant number 200140 from AFA Insurance and The Swedish Naprapathic Association.].  

4. In the online submission form, you indicated that [Due to ethical restrictions of disclosing personal data, authors have to seek permission to allow us to make the data used in this study available. Data will be available upon request after permission is granted from the Karolinska Institutet’s Ethics Review Board in Stockholm whose contact is kansli@stockholm.epn.se. Inquiries for data access should first be sent to iben.axen@ki.se, who will then contact the ethics board for permission to openly share the data.]. 

Additional Editor Comments:

Thank you for your submission, reviewers' reports are provided and both reviewers have commented on the quality of the work. They have also provided some suggestions for strengthening the work and you are invited to consider these as revisions to your manuscript.

Reviewers' comments:

Reviewer's Responses to Questions

**Comments to the Author**

1. Is the manuscript technically sound, and do the data support the conclusions?

Reviewer #1: Yes

Reviewer #2: Yes

2. Has the statistical analysis been performed appropriately and rigorously? 

Reviewer #1: I Don't Know

Reviewer #2: Yes

3. Have the authors made all data underlying the findings in their manuscript fully available?

Reviewer #1: No

Reviewer #2: Yes

4. Is the manuscript presented in an intelligible fashion and written in standard English?

Reviewer #1: Yes

Reviewer #2: Yes

5. Review Comments to the Author

Reviewer #1: Thank you for your manuscript and contribution to this body of knowledge. There are a few helpful details that would support your readers.

1. In the background, presenting specifics or examples of the voluntary recommendations would give context to the results.

2. A rationale for the use of a dichotomy when the original scale only had one 'no' options would prevent readers from questioning this decision.

Reviewer #2: The study is sound, robust in the business review, though there is a big opportunity being missed here (I think...). This data is a significant opportunity to compare to data from other areas, the USA, Australia, Continental Europe, Asia etc... What was business reduction in other countries in the same time period? Specific comparison to manual therapy only may not be able to made, but the economic downturn globally was massive. This study shows this Sweden it was minimal, controlled, and adaptable under less stringent government policy and the long term public health impact may have been to the better. Make the comparison as an additional component of the study. From the title there is the opportunity to show the limited impact in Sweden, not just the impact in Sweden without comparison.

6. PLOS authors have the option to publish the peer review history of their article (what does this mean?). If published, this will include your full peer review and any attached files.

Reviewer #1: No

Reviewer #2: No

---

## [Author Response · Author response to Decision Letter 0]

14 Jun 2024

Response to Reviewers’ comments

Thank you for considering our submission. The authors have gone through the reviewers' reports and have address the comments on the work which the authors believe has improved the quality of the work. The authors thank the reviewers for their suggestions for strengthening the work.

Reviewer #1: 

Thank you for your manuscript and contribution to this body of knowledge. There are a few helpful details that would support your readers.

1. In the background, presenting specifics or examples of the voluntary recommendations would give context to the results.

Thank you for the comment. The authors have presented specific examples of the voluntary recommendations in the introductory paragraphs to provide context to the results. 

‘‘Sweden's approach to COVID-19 differed from many other countries, as it did not enforce nationwide lockdowns and imposed fewer and less stringent restrictions. The response involved implementing various measures throughout the pandemic to control the virus's spread and lessen its impact on healthcare, individuals, and businesses. Sweden's strategy focused on non-binding restrictions, emphasizing personal responsibility through practices like social distancing, hand hygiene, and self-isolation when experiencing mild symptoms. Additional guidelines included remote work, virtual schooling, essential-travel-only policies, and limitations on public gatherings. The main authority overseeing Sweden's COVID-19 protocols was The Public Health Agency of Sweden, with regional directives also in place to manage local outbreaks, if needed.’’

2. A rationale for the use of a dichotomy when the original scale only had one 'no' options would prevent readers from questioning this decision.

The authors believe the reviewer is referencing the question asked to participants to comment on the following statement: “Official recommendations have interfered with clinical practice”, with the answer alternatives: “Not at all”, “Yes, to some degree”, “Yes, to a moderate degree”, “Yes, to a large degree”, and “I have not been clinically active during the COVID-19 pandemic”. The variable was later dichotomized by categorizing all the ‘‘yes-responses’’ into 1 = ‘‘yes’’ and “Not at all” to 0 = ‘‘no’’. 

The decision to dichotomize the variable is conceptually clear. All responses indicating any degree of interference with clinical practice ("Yes, to some degree", "Yes, to a moderate degree", "Yes, to a large degree") were grouped together as a "yes" response, indicating some level of disruption. The "Not at all" response was categorized as "no", indicating no disruption. All "yes" responses share the common characteristic of indicating disruption, which justifies treating them as a single group. This dichotomization simplifies the analysis and interpretation by focusing on the presence or absence of disruptions. 

The authors think that this needs to be addressed briefly in the discussions of potential limitations in the manuscript. 

Reviewer #2: 

The study is sound, robust in the business review, though there is a big opportunity being missed here (I think...). This data is a significant opportunity to compare to data from other areas, the USA, Australia, Continental Europe, Asia etc... What was business reduction in other countries in the same time period? Specific comparison to manual therapy only may not be able to made, but the economic downturn globally was massive. This study shows this Sweden it was minimal, controlled, and adaptable under less stringent government policy and the long term public health impact may have been to the better. Make the comparison as an additional component of the study. From the title there is the opportunity to show the limited impact in Sweden, not just the impact in Sweden without comparison.

Thank you for the comment. The authors agree with the reviewer that comparing the data to data from other areas including the USA, Australia, Continental Europe, Asia etc. concerning business reduction in the same period would have been better. However, that opportunity is limited for this study since the authors do have any other data from other countries or the continent. The authors have included a paragraph to highlight comparison with previous research.

‘‘The study is one of the few to investigate the economic impact of the COVID-19 policy restriction on businesses, and the strategies adopted by businesses and employers. Public health policies put in place during the COVID-19 pandemic have been assessed for their effectiveness in reducing transmission and minimizing health outcomes like death [21]. Limited research discusses the association between COVID-19 policy restrictions and economic impact, though not thoroughly examined. A descriptive study across eleven countries showed how chiropractors used innovative strategies such as telehealth and outreach to communicate and care for patients [12]. A global survey found that most chiropractors (85%) followed regulatory advice by using telehealth, personal protective equipment, and other measures in their practice [11]. Economic support such as wage and business subsidies were also mentioned, which were accessible in certain countries like Canada, Sweden, to help businesses and employers cope with costs and job losses. Although, the findings in the previous research are consistent with our findings in terms impact and strategies adopted by businesses and employers further research needed on global impact of manual therapists during pandemics. This information could advise government bodies and professional associations that manual therapists can implement a multifaceted strategy to provide financial support initiatives during future pandemics. These efforts have the capability to empower business owners, strengthen their resilience, and foster growth in the face of ongoing challenges.’’

---

## [Editor Report · Decision Letter 1]

2 Aug 2024

The impact of official recommendations during the COVID-19 pandemic on the clinical activity and business turnover of manual therapists in Sweden – the CAMP cohort study.

PONE-D-24-08211R1

Dear Dr. Aboagye,

We’re pleased to inform you that your manuscript has been judged scientifically suitable for publication and will be formally accepted for publication once it meets all outstanding technical requirements.

Kind regards,

Jenny Wilkinson, PhD

Academic Editor

PLOS ONE
---

## [Editor Report · Acceptance letter]

8 Aug 2024

PONE-D-24-08211R1 

PLOS ONE

Dear Dr. Aboagye, 

I'm pleased to inform you that your manuscript has been deemed suitable for publication in PLOS ONE. Congratulations! Your manuscript is now being handed over to our production team.

Kind regards, 

on behalf of

Dr Jenny Wilkinson 

Academic Editor

PLOS ONE